# Lifelong Learning with Non-i.i.d. Tasks

**Anastasia Pentina**
IST Austria
Klosterneuburg, Austria
apentina@ist.ac.at

**Christoph H. Lampert**
IST Austria
Klosterneuburg, Austria
chl@ist.ac.at

## Abstract

In this work we aim at extending the theoretical foundations of lifelong learning. Previous work analyzing this scenario is based on the assumption that learning tasks are sampled i.i.d. from a task environment or limited to strongly constrained data distributions. Instead, we study two scenarios when lifelong learning is possible, even though the observed tasks do not form an i.i.d. sample: first, when they are sampled from the same environment, but possibly with dependencies, and second, when the task environment is allowed to change over time in a consistent way. In the first case we prove a PAC-Bayesian theorem that can be seen as a direct generalization of the analogous previous result for the i.i.d. case. For the second scenario we propose to learn an inductive bias in form of a transfer procedure. We present a generalization bound and show on a toy example how it can be used to identify a beneficial transfer algorithm.

## 1 Introduction

Despite the tremendous growth of available data over the past decade, the lack of fully annotated data, which is an essential part of success of any traditional supervised learning algorithm, demands methods that allow good generalization from limited amounts of training data. One way to approach this is provided by the *lifelong learning* (or *learning to learn* [1]) paradigm, which is based on the idea of accumulating knowledge over the course of learning multiple tasks in order to improve the performance on future tasks.

In order for this scenario to make sense one has to define what kind of relations connect the observed tasks with the future ones. The first formal model of lifelong learning was proposed by Baxter in [2]. He introduced the notion of *task environment* – a set of all tasks that may need to be solved together with a probability distribution over them. In Baxter's model the lifelong learning system observes tasks that are sampled i.i.d. from the task environment. This allows proving bounds in the PAC framework [3, 4] that guarantee that a hypothesis set or inductive bias that works well on the observed tasks will also work well on future tasks from the same environment. Baxter's results were later extended using algorithmic stability [5], task similarity measures [6], and PAC-Bayesian analysis [7]. Specific cases that were studied include feature learning [8] and sparse coding [9].

All these works, however, assume that the observed tasks are independently and identically distributed, as the original work by Baxter did. This assumption allows making predictions about the future of the learning process, but it limits the applicability of the results in practice. To our knowledge, only the recent [10] has studied lifelong learning without an i.i.d. assumption. However, the considered framework is limited to binary classification with linearly separable classes and isotropic log-concave data distributions.

In this work we use the PAC-Bayesian framework to study two possible relaxations of the i.i.d. assumption without restricting the class of possible data distributions. First, we study the case in which tasks can have dependencies between them, but are still sampled from a fixed task environment. An

illustrative example would be when task are to predict the outcome of chess games. Whenever a player plays multiple games the corresponding tasks are not be independent. In this setting we retain many concepts of [7] and learn an inductive bias in the form of a probability distribution. We prove a bound relating the expected error when relying on the learned bias for future tasks to its empirical error over the observed tasks. It has the same form as for the i.i.d. situation, except for a slowdown of convergence proportional to a parameter capturing the amount of dependence between tasks.

Second, we introduce a new and more flexible lifelong learning setting, in which the learner observes a sequence of tasks from different task environments. This could be, e.g., classification tasks of increasing difficulty. In this setting one cannot expect that transferring an inductive bias from observed tasks to future tasks will be beneficial, because the task environment is not stationary. Instead, we aim at learning an effective *transfer algorithm*: a procedure that solves a task taking information from a previous task into account. We bound the expected performance of such algorithms when applied to future tasks based on their performance on the observed tasks.

## 2 Preliminaries

Following Baxter's model [2] we assume that all tasks that may need to be solved share the same input space $\mathcal{X}$ and output space $\mathcal{Y}$. The lifelong learning system observes $n$ tasks $t_1, \dots, t_n$ in form of training sets $S_1, \dots, S_n$, where each $S_i = \{(x_1^i, y_1^i), \dots, (x_m^i, y_m^i)\}$ is a set of $m$ points sampled i.i.d. from the corresponding unknown data distribution $D_i$ over $\mathcal{X} \times \mathcal{Y}$. In contrast to previous works on lifelong learning [2, 5, 8] we omit the assumption that the observed tasks are independently and identically distributed.

In order to theoretically analyze lifelong learning in the case of non-i.i.d. tasks we use techniques from PAC-Bayesian theory [11]. We assume that the learner uses the same hypothesis set $H = \{h : \mathcal{X} \to \mathcal{Y}\}$ and the same loss function $\ell : \mathcal{Y} \times \mathcal{Y} \to [0, 1]$ for solving all tasks. PAC-Bayesian theory studies the performance of randomized, Gibbs, predictors. Formally, for any probability distribution $Q$ over the hypothesis set, the corresponding Gibbs predictor for every point $x \in \mathcal{X}$ randomly samples $h \sim Q$ and returns $h(x)$. The expected loss of such Gibbs predictor on a task corresponding to a data distribution $D$ is given by:

$$\mathrm{er}(Q) = \mathbf{E}_{h \sim Q} \mathbf{E}_{(x,y) \sim D} \ell(h(x), y) \tag{1}$$

and its empirical counterpart based on a training set $S$ sampled from $D^m$ is given by:

$$\widehat{\mathrm{er}}(Q) = \mathbf{E}_{h \sim Q} \frac{1}{m} \sum_{i=1}^{m} \ell(h(x_i), y_i). \tag{2}$$

PAC-Bayesian theory allows us to obtain upper bounds on the difference between these two quantities of the following form:

**Theorem 1.** *Let $P$ be any distribution over $H$, fixed before observing the sample $S$. Then for any $\delta > 0$ the following holds uniformly for all distributions $Q$ over $H$ with probability at least $1 - \delta$:*

$$\mathrm{er}(Q) \leq \widehat{\mathrm{er}}(Q) + \frac{1}{\sqrt{m}} \mathrm{KL}(Q\|P) + \frac{1 + 8 \log 1/\delta}{8\sqrt{m}}, \tag{3}$$

*where $\mathrm{KL}$ denotes the Kullback-Leibler divergence.*

The distribution $P$ should be chosen before observing any data and therefore is usually referred as *prior* distribution. In contrast, the bound holds uniformly with respect to the distributions $Q$. Whenever it consists only of computable quantities, it can be used to choose a data-dependent $Q$ that minimizes the right hand side of the inequality (3) and thus provides a Gibbs predictor with expected error bounded by a hopefully low value. Suchwise $Q$ is usually referred as a *posterior* distribution. Note that besides *explicit* bounds, such as (3), in the case of $0/1$-loss one can also derive *implicit* bound that can be tighter in some regimes [12]. Instead of the error difference, $\mathrm{er} - \widehat{\mathrm{er}}$, these bound their KL-divergence, $kl(\widehat{\mathrm{er}} \| \mathrm{er})$, where $kl(q\|p)$ denotes the KL-divergence between two Bernoulli random variables with success probabilities $p$ and $q$. In this work, we prefer explicit bounds as they are more intuitive and allow for more freedom in the choice of different loss functions. They also allow us to combine several inequalities in an additive way, which we make use of in Sections 3 and 4.

# 3 Dependent tasks

The first extension of Baxter's model that we study is the case, when the observed tasks are sampled from the same task environment, but with some interdependencies. In other words, in this case the tasks are identically, but not independently, distributed.

Since the task environment is assumed to be constant we can build on ideas from the situation of i.i.d. tasks in [7]. We assume that for all tasks the learner uses the same deterministic learning algorithm that produces a posterior distribution $Q$ based on a prior distribution $P$ and a sample set $S$. We also assume that there is a set of possible prior distributions and some hyper-prior distribution $\mathcal{P}$ over it. The goal of the learner is to find a hyper-posterior distribution $\mathcal{Q}$ over this set such that, when the prior is sampled according to $\mathcal{Q}$, the expected loss on the next, yet unobserved task is minimized:

$$\mathrm{er}(\mathcal{Q}) = \mathbf{E}_{P\sim\mathcal{Q}}\mathbf{E}_{(t,S_t)}\mathbf{E}_{h\sim Q(P,S_t)}\mathbf{E}_{(x,y)\sim D_t}\ell(h(x),y). \tag{4}$$

The empirical counterpart of the above quantity is given by:

$$\widehat{\mathrm{er}}(\mathcal{Q}) = \mathbf{E}_{P\sim\mathcal{Q}}\frac{1}{n}\sum_{i=1}^{n}\mathbf{E}_{h\sim Q_i(P,S_i)}\frac{1}{m}\sum_{j=1}^{m}\ell(h(x_j^i),y_j^i). \tag{5}$$

In order to bound the difference between these two quantities we adopt the two-staged procedure used in [7]. First, we bound the difference between the empirical error $\widehat{\mathrm{er}}(\mathcal{Q})$ and the corresponding expected multi-task risk given by:

$$\widetilde{\mathrm{er}}(\mathcal{Q}) = \mathbf{E}_{P\sim\mathcal{Q}}\frac{1}{n}\sum_{i=1}^{n}\mathbf{E}_{h\sim Q_i(P,S_i)}\mathbf{E}_{(x,y)\sim D_i}\ell(h(x),y). \tag{6}$$

Then we continue with bounding the difference between $\mathrm{er}(\mathcal{Q})$ and $\widetilde{\mathrm{er}}(\mathcal{Q})$.

Since conditioned on the observed tasks the corresponding training samples are independent, we can reuse the following results from [7] in order to perform the first step of the proof.

**Theorem 2.** *With probability at least $1-\delta$ uniformly for all $\mathcal{Q}$:*

$$\widetilde{\mathrm{er}}(\mathcal{Q}) \leq \widehat{\mathrm{er}}(\mathcal{Q}) + \frac{1}{n\sqrt{m}}\Big(\mathrm{KL}(\mathcal{Q}||\mathcal{P}) + \sum_{i=1}^{n}\mathbf{E}_{P\sim\mathcal{Q}}\mathrm{KL}(Q_i(P,S_i)||P)\Big) + \frac{n+8\log(1/\delta)}{8n\sqrt{m}}. \tag{7}$$

To bound the difference between $\mathrm{er}(\mathcal{Q})$ and $\widetilde{\mathrm{er}}(\mathcal{Q})$, however, the results from [7] cannot be used, because they rely on the assumption that the observed tasks are independent. Instead we adopt ideas from chromatic PAC-Bayesian bounds [13] that rely on the properties of a dependency graph built with respect to the dependencies within the observed tasks.

**Definition 1** (Dependency graph). *The dependency graph $\Gamma(\mathbf{t}) = (V,E)$ of a set of random variables $\mathbf{t} = (t_1,\ldots,t_n)$ is such that:*

- *the set of vertices $V$ equals $\{1,\ldots,n\}$,*

- *there is no edge between $i$ and $j$ if and only if $t_i$ and $t_j$ are independent.*

**Definition 2** (Exact fractional cover [14]). *Let $\Gamma = (V,E)$ be an undirected graph with $V = \{1,\ldots,n\}$. A set $\mathbf{C} = \{(C_j,w_j)\}_{j=1}^{k}$, where $C_j \subset V$ and $w_j \in [0,1]$ for all $j$, is a proper exact fractional cover if:*

- *for every $j$ all vertices in $C_j$ are independent,*

- *$\cup_j C_j = V$,*

- *for every $i \in V$ $\quad \sum_{j=1}^{k} w_j \mathbb{I}_{i\in C_j} = 1$.*

*The sum of the weights $\mathbf{w}(\mathbf{C}) = \sum_{j=1}^{k} w_j$ is the chromatic weight of $\mathbf{C}$ and $k$ is the size of $\mathbf{C}$.*

Then the following holds:

**Theorem 3.** *For any fixed hyper-prior distribution $\mathcal{P}$, any proper exact fractional cover $\mathbf{C}$ of the dependency graph $\Gamma(t_1, \ldots, t_n)$ of size $k$ and any $\delta > 0$ the following holds with probability at least $1 - \delta$ uniformly for all hyper-posterior distributions $\mathcal{Q}$:*

$$\mathrm{er}(\mathcal{Q}) \leq \widetilde{\mathrm{er}}(\mathcal{Q}) + \sqrt{\frac{\mathbf{w}(\mathbf{C})}{n}} \, \mathrm{KL}(\mathcal{Q}\|\mathcal{P}) + \frac{\sqrt{\mathbf{w}(\mathbf{C})}(1 - 8\log\delta + 8\log k)}{8\sqrt{n}}. \tag{8}$$

*Proof.* By Donsker-Varadhan's variational formula [15]:

$$\mathrm{er}(\mathcal{Q}) - \widetilde{\mathrm{er}}(\mathcal{Q}) = \sum_{j=1}^{k} \frac{w_j}{\mathbf{w}(\mathbf{C})} \mathbf{E}_{P \sim \mathcal{Q}} \frac{\mathbf{w}(\mathbf{C})}{n} \sum_{i \in C_j} \mathbf{E}_{(t,S_t)} \mathrm{er}_t(Q_t) - \mathrm{er}_i(Q_i) \leq \tag{9}$$

$$\sum_{j=1}^{k} \frac{w_j}{\mathbf{w}(\mathbf{C})} \frac{1}{\lambda_j} \left( \mathrm{KL}(\mathcal{Q}\|\mathcal{P}) + \log \mathbf{E}_{P \sim \mathcal{P}} \exp \left( \frac{\lambda_j \mathbf{w}(\mathbf{C})}{n} \sum_{i \in C_j} \mathbf{E}_{(t,S_t)} \mathrm{er}_t(Q_t) - \mathrm{er}_i(Q_i) \right) \right).$$

Since the tasks within every $C_j$ are independent, for every fixed prior $P$ $\{\mathbf{E}_{(t,S_t)} \mathrm{er}_t(Q_t) - \mathrm{er}_i(Q_i)\}_{i \in C_j}$ are i.i.d. and take values in $[b - 1, b]$, where $b = \mathbf{E}_{(t,S_t)} \mathrm{er}_t(Q_t)$. Therefore, by Hoeffding's lemma [16]:

$$\mathbf{E}_{(t_i,S_i), i \in C_j} \exp \left( \frac{\lambda_j \mathbf{w}(\mathbf{C})}{n} \sum_{i \in C_j} \mathbf{E}_{(t,S_t)} \mathrm{er}_t(Q_t) - \mathrm{er}_i(Q_i) \right) \leq \exp \left( \frac{\lambda_j^2 \mathbf{w}(\mathbf{C})^2 |C_j|}{8n^2} \right). \tag{10}$$

Therefore, by Markov's inequality with probability at least $1 - \delta_j$ it holds that:

$$\log \mathbf{E}_{P \sim \mathcal{P}} \exp \left( \frac{\lambda_j \mathbf{w}(\mathbf{C})}{n} \sum_{i \in C_j} \mathbf{E}_{(t,S_t)} \mathrm{er}_t(Q_t) - \mathrm{er}_i(Q_i) \right) \leq \frac{\lambda_j^2 \mathbf{w}(\mathbf{C})^2 |C_j|}{8n^2} - \log \delta_j. \tag{11}$$

Consequently, we obtain with probability at least $1 - \sum_{j=1}^{k} \delta_j$:

$$\mathrm{er}(\mathcal{Q}) - \widetilde{\mathrm{er}}(\mathcal{Q}) \leq \sum_{j=1}^{k} \frac{w_j}{\mathbf{w}(\mathbf{C})} \frac{1}{\lambda_j} \mathrm{KL}(\mathcal{Q}\|\mathcal{P}) + \sum_{j=1}^{k} \frac{w_j \lambda_j \mathbf{w}(\mathbf{C}) |C_j|}{8n^2} - \sum_{j=1}^{k} \frac{w_j}{\mathbf{w}(\mathbf{C}) \lambda_j} \log \delta_j. \tag{12}$$

By setting $\lambda_1 = \cdots = \lambda_k = \sqrt{n/\mathbf{w}(\mathbf{C})}$ and $\delta_j = \delta/k$ we obtain the statement of the theorem. $\square$

By combining Theorems 2 and 3 we obtain the main result of this section:

**Theorem 4.** *For any fixed hyper-prior distribution $\mathcal{P}$, any proper exact fractional cover $\mathbf{C}$ of the dependency graph $\Gamma(t_1, \ldots, t_n)$ of size $k$ and any $\delta > 0$ the following holds with probability at least $1 - \delta$ uniformly for all hyper-posterior distributions $\mathcal{Q}$:*

$$\mathrm{er}(\mathcal{Q}) \leq \widehat{\mathrm{er}}(\mathcal{Q}) + \frac{1 + \sqrt{\mathbf{w}(\mathbf{C})mn}}{n\sqrt{m}} \mathrm{KL}(\mathcal{Q}\|\mathcal{P}) + \frac{1}{n\sqrt{m}} \sum_{i=1}^{n} \mathbf{E}_{P \sim \mathcal{Q}} \mathrm{KL}(Q_i(P, S_i)\|P) +$$

$$\frac{n + 8\log(2/\delta)}{8n\sqrt{m}} + \frac{\sqrt{\mathbf{w}(\mathbf{C})}(1 + 8\log(2/\delta) + 8\log k)}{8\sqrt{n}}. \tag{13}$$

Theorem 4 shows that even in the case of non-independent tasks a bound very similar to that in [7] can be obtained. In particular, it contains two types of complexity terms: $\mathrm{KL}(\mathcal{Q}\|\mathcal{P})$ corresponds to the level of the task environment and $\mathrm{KL}(Q_i\|P)$ corresponds specifically to the $i$-th task. Similarly to the i.i.d. case, when the learner has access to unlimited amount of data, but for finitely many observed tasks ($m \to \infty$, $n < \infty$), the complexity terms of the second type converge to 0 as $1/\sqrt{m}$, while the first one does not, as there is still uncertainty on the task environment level. In the opposite situation, when the learner has access to infinitely many tasks, but with only finitely many samples per task ($m < \infty$, $n \to \infty$), the first complexity term converges to 0 as $\sqrt{\mathbf{w}(\mathbf{C})/n}$, up to logarithmic terms. Thus there is a worsening comparing to the i.i.d. case proportional to $\sqrt{\mathbf{w}(\mathbf{C})}$, which represents the amount of dependence among the tasks. If the tasks are actually i.i.d., the dependency graph contains no edges, so we can form a cover of size 1 with chromatic weight 1. Thus we recover the result from [7] as a special case of Theorem 4.

For general dependence graph, fastest convergence is obtained by using a cover with minimal chromatic weight. It is known that the minimal chromatic weight, $\chi^*(\Gamma)$, satisfies the following inequality [14]: $1 \leq c(\Gamma) \leq \chi^*(\Gamma) \leq \Delta(\Gamma) + 1$, where $c(\Gamma)$ is the order of the largest clique in $\Gamma$ and $\Delta(\Gamma)$ is the maximum degree of a vertex in $\Gamma$.

In some situations, even the bound obtainable from Theorem 4 by plugging in a cover with the minimal chromatic weight can be improved: Theorem 4 also holds for any subset $\mathbf{t}_s$, $|\mathbf{t}_s| = s$, of the observed tasks with the induced dependency subgraph $\Gamma_s$. Therefore it might provide a tighter bound if $\chi^*(\Gamma_s)/s$ is smaller than $\chi^*(\Gamma)/n$. However, this is not guaranteed since the empirical error $\widehat{\mathrm{er}}$ computed on $\mathbf{t}_s$ might become larger, as well as the second part of the bound, which decreases with $n$ and does not depend on the chromatic weight of the cover. Note also that such a subset needs to be chosen before observing the data, since the bound of Theorem 4 holds with probability $1 - \delta$ only for a fixed set of tasks and a fixed cover.

Another important aspect of Theorem 4 as a PAC-Bayesian bound is that the right hand side of inequality (13) consists only of computable quantities. Therefore it can be seen as quality measure of a hyper-posterior $\mathcal{Q}$ and by minimizing it one could obtain a distribution that is adjusted to a particular task environment. The resulting minimizer can be expected to work well even on new, yet unobserved tasks, because the guarantees of Theorem 4 still hold due to the uniformity of the bound. To do so, one can use the same techniques as in [7], because Theorem 4 differs from the bound provided there only by constant factors.

## 4    Changing Task Environments

In this section we study a situation, when the task environment is gradually changing: every next task $t_{i+1}$ is sampled from a distribution $T_{i+1}$ over the tasks that can depend on the history of the process. Due to the change of task environment the previous idea of learning one prior for all tasks does not seem reasonable anymore. In contrast, we propose to learn a transfer algorithm that produces a solution for the current task based on the corresponding sample set and the sample set from the previous task. Formally, we assume that there is a set $\mathcal{A}$ of learning algorithms that produce a posterior distribution $Q_{i+1}$ for task $t_{i+1}$ based on the training samples $S_i$ and $S_{i+1}$. The goal of the learner is to identify an algorithm $A$ in this set that leads to good performance when applied to a new, yet unobserved task, while using the last observed training sample $S_n$[1].

For each task $t_i$ and each algorithm $A \in \mathcal{A}$ we define the expected and empirical error of applying this algorithm as follows:

$$\mathrm{er}_i(A) = \mathbf{E}_{h \sim Q_i} \mathbf{E}_{(x,y) \sim D_i} \ell(h(x), y), \quad \widehat{\mathrm{er}}_i(A) = \mathbf{E}_{h \sim Q_i} \frac{1}{m} \sum_{j=1}^m \ell(h(x_j^i), y_j^i), \qquad (14)$$

where $Q_i = A(S_i, S_{i-1})$. The goal of the learner is to find $A$ that minimizes $\mathrm{er}_{n+1}$ given the history of the observed tasks. However, if the task environment would change arbitrarily from step to step, the observed tasks would not contain any relevant information for a new task. To overcome this difficulty, we make the assumption that the expected performance of the algorithms in $\mathcal{A}$ does not change over time. Formally, we assume for each $A \in \mathcal{A}$ there exists a value, $\mathrm{er}(A)$, such that for every $i = 2, \ldots, n+1$, with $E_i = (T_i, t_i, S_i)$:

$$\mathbf{E}_{\{E_{i-1}, E_i\}}[\mathrm{er}_i(A) \,|\, E_1, \ldots, E_{i-2}] = \mathrm{er}(A). \qquad (15)$$

In words, the quality of a transfer algorithm does not depend on when during the task sequence it is applied, provided that it is always applied to the subsequent sample sets. Note that this is a natural assumption for lifelong learning: without it, the quality of transfer algorithms could change over time, so an algorithm that works well for all observed tasks might not work anymore for future tasks.

The goal of the learner can be reformulated as identifying $A \in \mathcal{A}$ with minimal $\mathrm{er}(A)$, which can be seen as the expected value of the expected risk of applying algorithm $A$ on the next, yet unobserved task. Since $\mathrm{er}(A)$ is unknown, we derive an upper bound based on the observed data that holds uniformly for all algorithms $A$ and therefore can be used to guide the learner. To do so, we again use

the construction with hyper-priors and hyper-posteriors from the previous section. Formally, let $\mathcal{P}$ be a prior distribution over the set of possible algorithms that is fixed before any data arrives and let $\mathcal{Q}$ be a possibly data-dependent hyper-posterior. The quality of the hyper-posterior and its empirical counterpart are given by the following quantities:

$$\text{er}(\mathcal{Q}) = \mathbf{E}_{A\sim\mathcal{Q}} \text{er}(A), \qquad \widehat{\text{er}}(\mathcal{Q}) = \mathbf{E}_{A\sim\mathcal{Q}} \frac{1}{n-1}\sum_{i=2}^{n} \widehat{\text{er}}_i(A). \tag{16}$$

Similarly to the previous section, we first bound the difference between $\widehat{\text{er}}(\mathcal{Q})$ and multi-task expected error given by:

$$\widetilde{\text{er}}(\mathcal{Q}) = \mathbf{E}_{A\sim\mathcal{Q}} \frac{1}{n-1}\sum_{i=2}^{n} \text{er}_i(A). \tag{17}$$

Even though Theorem 2 is not directly applicable here, a more careful modification of it allows to obtain the following result (see supplementary material for a detailed proof):

**Theorem 5.** *For any fixed hyper-prior distribution $\mathcal{P}$ with probability at least $1-\delta$ the following holds uniformly for all hyper-posterior distributions $\mathcal{Q}$:*

$$\widetilde{\text{er}}(\mathcal{Q}) \leq \widehat{\text{er}}(\mathcal{Q}) + \frac{1}{(n-1)\sqrt{m}} \text{KL}(\mathcal{Q}\times Q_2\times\cdots\times Q_n || \mathcal{P}\times P_2\times\cdots\times P_n) + \frac{(n-1)+8\log(1/\delta)}{8(n-1)\sqrt{m}},$$

*where $P_2,\ldots,P_n$ are some reference prior distributions that do not depend on the training sets of subsequent tasks. Possible choices include using just one prior distribution $P$ fixed before observing any data, or using the posterior distributions obtained from the previous task, i.e. $P_i = Q_{i-1}$.*

To complete the proof we need to bound the difference between $\text{er}(\mathcal{Q})$ and $\widetilde{\text{er}}(\mathcal{Q})$. We use techniques from [17] in combination of those from [13], resulting in the following lemma:

**Lemma 1.** *For any fixed algorithm $A$ and any $\lambda$ the following holds:*

$$\mathbf{E}_{E_1,\ldots,E_n} \exp\left(\lambda\left(\text{er}(A) - \frac{1}{n-1}\sum_{i=2}^{n} \text{er}_i(A)\right)\right) \leq \exp\left(\frac{\lambda^2}{2(n-1)}\right). \tag{18}$$

*Proof.* First, define $X_i = (E_{i-1}, E_i)$ for $i = 2,\ldots,n$ and $g : X_i \mapsto \text{er}_i(A)$ and $b = \text{er}(A)$. Then:

$$\exp\left(\lambda\left(\text{er}(A) - \frac{1}{n-1}\sum_{i=2}^{n} \text{er}_i(A)\right)\right) = \exp\left(\frac{\lambda}{n-1}\left(\sum_{\text{even } i}(b - g(X_i)) + \sum_{\text{odd } i}(b - g(X_i))\right)\right)$$

$$\leq \frac{1}{2}\exp\left(\frac{2\lambda}{n-1}\sum_{\text{even } i}(b - g(X_i))\right) + \frac{1}{2}\exp\left(\frac{2\lambda}{n-1}\sum_{\text{odd } i}(b - g(X_i))\right). \tag{19}$$

Note, that both, the set of $X_i$-s corresponding to even $i$ and the set of $X_i$-s corresponding to odd $i$, form a martingale difference sequence. Therefore by using Lemma 2 from the supplementary material (or similarly Lemma 2 in [17]) and Hoeffding's lemma [16] we obtain:

$$\mathbf{E}_{E_1,\ldots,E_n} \exp\left(\frac{2\lambda}{n-1}\sum_{\text{even } i}(b - g(X_i))\right) \leq \exp\left(\frac{4\lambda^2}{8(n-1)}\right) \tag{20}$$

and the same for the odd $i$. Together with inequality (19) it gives the statement of the lemma. $\qquad\square$

Now we can prove the following statement:

**Theorem 6.** *For any hyper-prior distribution $\mathcal{P}$ and any $\delta > 0$ with probability at least $1-\delta$ the following inequality holds uniformly for all $\mathcal{Q}$:*

$$\text{er}(\mathcal{Q}) \leq \widetilde{\text{er}}(\mathcal{Q}) + \frac{1}{\sqrt{n-1}} \text{KL}(\mathcal{Q}||\mathcal{P}) + \frac{1+2\log(1/\delta)}{2\sqrt{n-1}}. \tag{21}$$

*Proof.* By applying Donsker-Varadhan's variational formula [15] one obtains that:

$$\text{er}(\mathcal{Q}) - \widetilde{\text{er}}(\mathcal{Q}) \leq \frac{1}{\lambda}\left(\text{KL}(\mathcal{Q}||\mathcal{P}) + \log \mathbf{E}_{A\sim\mathcal{P}} \exp\lambda\left(\text{er}(A) - \frac{1}{n-1}\sum_{i=2}^{n} \text{er}_i(A)\right)\right). \tag{22}$$

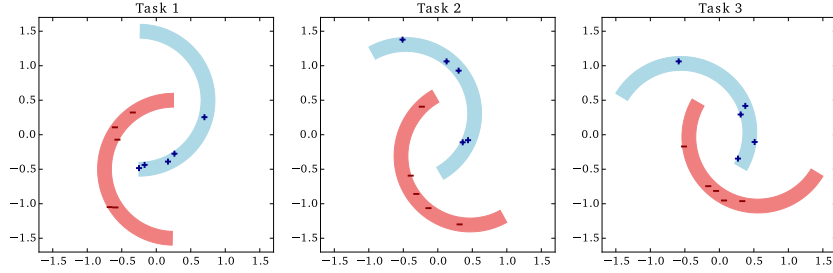

Figure 1: Illustration of three learning tasks sampled from a non-stationary environment. Shaded areas illustrate the data distribution, $+$ and $-$ indicate positive and negative training examples. Between subsequent tasks, the data distribution changes by a rotation. A transfer algorithm with access to two subsequent tasks can compensate for this by rotating the previous data into the new position, thereby obtaining more data samples to train on.

For a fixed algorithm $A$ we obtain from Lemma 1:

$$\mathbf{E}_{E_1,\ldots,E_n} \exp\left(\lambda\Big(\mathrm{er}(A) - \frac{1}{n-1}\sum_{i=2}^{n}\mathrm{er}_i(A)\Big)\right) \le \exp\left(\frac{\lambda^2}{2(n-1)}\right). \tag{23}$$

Since $\mathcal{P}$ does not depend on the process, by Markov's inequality, with probability at least $1-\delta$, we obtain

$$\mathbf{E}_{A\sim\mathcal{P}} \exp\lambda\left(\mathrm{er}(A) - \frac{1}{n-1}\sum_{i=2}^{n}\mathrm{er}_i(A)\right) \le \frac{1}{\delta}\exp\left(\frac{\lambda^2}{2(n-1)}\right). \tag{24}$$

The statement of the theorem follows by setting $\lambda = \sqrt{n-1}$. $\qquad\square$

By combining Theorems 5 and 6 we obtain the main result of this section:

**Theorem 7.** *For any hyper-prior distribution $\mathcal{P}$ and any $\delta > 0$ with probability at least $1-\delta$ the following holds uniformly for all $\mathcal{Q}$:*

$$\mathrm{er}(\mathcal{Q}) \le \widehat{\mathrm{er}}(\mathcal{Q}) + \frac{\sqrt{(n-1)m}+1}{(n-1)\sqrt{m}}\,\mathrm{KL}(\mathcal{Q}\|\mathcal{P}) + \frac{1}{(n-1)\sqrt{m}}\sum_{i=2}^{n}\mathbf{E}_{A\sim\mathcal{Q}}\,\mathrm{KL}(Q_i\|P_i)$$

$$+ \frac{(n-1)+8\log(2/\delta)}{8(n-1)\sqrt{m}} + \frac{1+2\log(2/\delta)}{2\sqrt{n-1}}, \tag{25}$$

*where $P_2,\ldots,P_n$ are some reference prior distributions that should not depend on the data of subsequent tasks.*

Similarly to Theorem 4 the above bound contains two types of complexity terms: one corresponding to the level of the changes in the task environment and task-specific terms. The first complexity term converges to 0 like $1/\sqrt{n-1}$ when the number of the observed tasks increases, indicating that more observed tasks allow for better estimation of the behavior of the transfer algorithms. The task-specific complexity terms vanish only when the amount of observed data $m$ per tasks grows. In addition, since the right hand side of the inequality (25) consists only of computable quantities and at the same time holds uniformly for all $\mathcal{Q}$, one can obtain a posterior distribution by minimizing it over the transfer algorithms that is adjusted to particularly changing task environments.

We illustrate this process by discussing a toy example (Figure 1). Suppose that $\mathcal{X} = \mathbb{R}^2$, $\mathcal{Y} = \{-1,1\}$ and that the learner uses linear classifiers, $h(x) = \mathrm{sign}\langle w, x\rangle$, and 0/1-loss, $\ell(y_1, y_2) = [\![y_1 \ne y_2]\!]$, for solving every task. For simplicity we assume that every task environment contains only one task or, equivalently, every $T_i$ is a delta peak, and that the change in the environment between two steps is due to a constant rotation by $\theta_0 = \frac{\pi}{6}$ of the feature space. For the set $\mathcal{A}$ we use a one-parameter family of transfer algorithms, $A_\alpha$ for $\alpha \in \mathbb{R}$. Given sample sets $S_{\mathrm{prev}}$ and $S_{\mathrm{cur}}$, any algorithm $A_\alpha$ first rotates $S_{\mathrm{prev}}$ by the angle $\alpha$, and then trains a linear support vector machine on the union of both sets. Clearly, the quality of each transfer algorithm depends on the chosen angle, and an elementary calculation shows that condition (15) is fulfilled. We can therefore use the bound (25)

as a criterion to determine a beneficial angle[2]. For that we set $Q_i = \mathcal{N}(w_i, I_2)$, i.e. unit variance Gaussian distributions with means $w_i$. Similarly, we choose all reference prior distributions as unit variance Gaussian with zero mean, $P_i = \mathcal{N}(0, I_2)$. Analogously, we set the hyper-prior $\mathcal{P}$ to be $\mathcal{N}(0, 10)$, a zero mean normal distribution with enlarged variance in order to make all reasonable rotations $\alpha$ lie within one standard deviation from the mean. As hyper-posteriors $\mathcal{Q}$ we choose $\mathcal{N}(\theta, 1)$ and the goal of the learning is to identify the best $\theta$. In order to obtain the objective function from equation (25) we first compute the complexity terms (and approximate all expectations with respect to $\mathcal{Q}$ by the values at its mean $\theta$):

$$\mathrm{KL}(\mathcal{Q}||\mathcal{P}) = \frac{\theta^2}{20}, \qquad \mathbf{E}_{A \sim \mathcal{Q}} \mathrm{KL}(Q_i||P_i) \approx \frac{\|w_i\|^2}{2}.$$

The empirical error of the Gibbs classifiers in the case of $0/1$-loss and Gaussian distributions is given by the following expression (we again approximate the expectation by the value at $\theta$) [20, 21]:

$$\widehat{\mathrm{er}}(\mathcal{Q}) \approx \frac{1}{n-1} \sum_{i=2}^{n} \frac{1}{m} \sum_{j=1}^{m} \overline{\Phi}\left(\frac{y_j^i \langle w_i, x_j^i \rangle}{\|x_j^i\|}\right), \tag{26}$$

where $\overline{\Phi}(z) = \frac{1}{2}\left(1 - \mathrm{erf}(\frac{z}{\sqrt{2}})\right)$ and $\mathrm{erf}(z)$ is the Gauss error function. The resulting objective function that we obtain for identifying a beneficial angle $\theta$ is the following:

$$\mathcal{J}(\theta) = \frac{\sqrt{(n-1)m} + 1}{(n-1)\sqrt{m}} \cdot \frac{\theta^2}{20} + \frac{1}{n-1} \sum_{i=2}^{n} \left(\frac{\|w_i\|^2}{2\sqrt{m}} + \frac{1}{m} \sum_{j=1}^{m} \overline{\Phi}\left(\frac{y_j^i \langle w_i, x_j^i \rangle}{\|x_j^i\|}\right)\right). \tag{27}$$

Numeric experiments confirm that by optimizing $\mathcal{J}(\theta)$ with respect to $\theta$ one can obtain an advantageous angle: using $n = 2, \dots, 11$ tasks, each with $m = 10$ samples, we obtain an average test error of $14.2\%$ for the $(n+1)$th task. As can be expected, this lies in between the error for the same setting without transfer, which was $15.0\%$, and the error when always rotating by $\frac{\pi}{6}$, which was $13.5\%$.

## 5   Conclusion

In this work we present a PAC-Bayesian analysis of lifelong learning under two types of relaxations of the i.i.d. assumption on the tasks. Our results show that accumulating knowledge over the course of learning multiple tasks can be beneficial for the future even if these tasks are not i.i.d. In particular, for the situation when the observed tasks are sampled from the same task environment but with possible dependencies we prove a theorem that generalizes the existing bound for the i.i.d. case. As a second setting we further relax the i.i.d. assumption and allow the task environment to change over time. Our bound shows that it is possible to estimate the performance of applying a transfer algorithm on future tasks based on its performance on the observed ones. Furthermore, our result can be used to identify a beneficial algorithm based on the given data and we illustrate this process on a toy example. For future work, we plan to expand on this aspect. Essentially, any existing domain adaptation algorithm can be used as a transfer method in our setting. However, the success of domain adaptation techniques is often caused by asymmetry between the source and the target - such algorithms usually rely on availability of extensive amounts of data from the source and only limited amounts from the target. In contrast, in lifelong learning setting all the tasks are assumed to be equipped with limited training data. Therefore we are particularly interested in identifying how far the constant quality assumption can be carried over to existing domain adaptation techniques and real-world lifelong learning situations.

**Acknowledgments.**   This work was in parts funded by the European Research Council under the European Union's Seventh Framework Programme (FP7/2007-2013)/ERC grant agreement no 308036.

## Footnotes

[1] Note that this setup includes the possibility of *model selection*, such as predictors using different feature representations or (hyper)parameter values.

[2]Note that Theorem 7 provides an upper bound for the expected error of stochastic Gibbs classifiers, and not deterministic ones that are preferable in practice. However for $0/1$-loss the error of a Gibbs classifier is bounded from below by half the error of the corresponding majority vote predictor [18, 19] and therefore twice the right hand side of (25) provides a bound for deterministic classifiers.

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
