[Supplementary Material · supplementary.pdf]

# Lifelong Learning with Non-i.i.d. Tasks
# Supplementary material

**Anastasia Pentina**
IST Austria
Klosterneuburg, Austria
apentina@ist.ac.at

**Christoph H. Lampert**
IST Austria
Klosterneuburg, Austria
chl@ist.ac.at

**Theorem 1.** *Let $P$ be any distribution over $H$, fixed before observing the sample $S$. Then for any $\delta > 0$ the following holds uniformly for all distributions $Q$ over $H$ with probability at least $1 - \delta$:*

$$\mathrm{er}(Q) \leq \widehat{\mathrm{er}}(Q) + \frac{1}{\sqrt{m}} \mathrm{KL}(Q||P) + \frac{1 + 8\log(1/\delta)}{8\sqrt{m}}. \tag{1}$$

*Proof.* Let $f(Q) = \mathrm{er}(Q) - \widehat{\mathrm{er}}(Q) = \mathbf{E}_{h \sim Q}\left(\mathbf{E}_{(x,y) \sim D}\ell(h(x), y) - \frac{1}{m}\sum_{i=1}^{m}\ell(h(x_i), y_i)\right)$. From Donsker-Varadhan's variational formula one obtains that for any $\lambda > 0$:

$$f(Q) \leq \frac{1}{\lambda}\left(\mathrm{KL}(Q||P) + \log \mathbf{E}_{h \sim P} \exp \lambda \left(\mathbf{E}_{(x,y) \sim D}\ell(h(x), y) - \frac{1}{m}\sum_{i=1}^{m}\ell(h(x_i), y_i)\right)\right). \tag{2}$$

Since loss function is bounded by 1, from Hoeffding's lemma we know that:

$$\mathbf{E}_{(x_i,y_i) \sim D} \exp\left(\frac{\lambda}{m}(\mathbf{E}_{(x,y) \sim D}\ell(h(x), y) - \ell(h(x_i), y_i))\right) \leq \exp\left(\frac{\lambda^2}{8m^2}\right). \tag{3}$$

Because the sample points are i.i.d., we can obtain that:

$$\mathbf{E}_{S \sim D^m} \exp \lambda \left(\mathbf{E}_{(x,y) \sim D}\ell(h(x), y) - \frac{1}{m}\sum_{i=1}^{m}\ell(h(x_i), y_i)\right) \leq \exp\left(\frac{\lambda^2}{8m}\right). \tag{4}$$

By combining the fact, that $P$ doesn't depend on $S$, and Markov's inequality, we obtain that with probability at least $1 - \delta$:

$$\mathbf{E}_{h \sim P} \exp \lambda \left(\mathbf{E}_{(x,y) \sim D}\ell(h(x), y) - \frac{1}{m}\sum_{i=1}^{m}\ell(h(x_i), y_i)\right) \leq \frac{1}{\delta} \exp\left(\frac{\lambda^2}{8m}\right). \tag{5}$$

By plugging it into (2) and setting $\lambda = \sqrt{m}$ we obtain the statement of the theorem. $\square$

**Lemma 2.** *Let $X_1, \ldots, X_n \in \Omega$ be a sequence of random variables and $g : \Omega \to [0, 1]$ be a function such that $\mathbf{E}[g(X_i)|X_1, \ldots, X_{i-1}] = b_i$. Let $Z_1, \ldots, Z_n$ be independent Bernoulli random variables such that $\mathbf{E}[Z_i] = b_i$. Then for any convex function $f$:*

$$\mathbf{E}[f(g(X_1), \ldots, g(X_n))] \leq \mathbf{E}[f(Z_1, \ldots, Z_n)]. \tag{6}$$

*Proof.* Any point $x = (x_1, \ldots, x_n) \in [0, 1]^n$ can be written as a linear combination of the extreme points $\nu = (\nu_1, \ldots, \nu_n) \in \{0, 1\}^n$ in the following way:

$$x = \sum_{\nu \in \{0,1\}^n} \left(\prod_{i=1}^{n}((1 - x_i)(1 - \nu_i) + x_i\nu_i)\right) \nu. \tag{7}$$

Therefore by convexity of $f$ we have that:

$$f(x) \leq \sum_{\nu \in \{0,1\}^n} \left( \prod_{i=1}^{n} ((1-x_i)(1-\nu_i) + x_i \nu_i) \right) f(\nu). \tag{8}$$

By taking expectations on both sides we obtain that:

$$\mathbf{E}_{X_1^n} f(g(X_1), \ldots, g(X_n)) \leq$$

$$\mathbf{E}_{X_1^n} \left[ \sum_{\nu \in \{0,1\}^n} \left( \prod_{i=1}^{n} ((1-g(X_i))(1-\nu_i) + g(X_i)\nu_i) \right) f(\nu) \right] =$$

$$\sum_{\nu \in \{0,1\}^n} \mathbf{E}_{X_1^n} \left[ \prod_{i=1}^{n} ((1-g(X_i))(1-\nu_i) + g(X_i)\nu_i) \right] f(\nu) =$$

$$\sum_{\nu \in \{0,1\}^n} \mathbf{E}_{X_1^{n-1}} \left[ \mathbf{E}_{X_n} \left[ \prod_{i=1}^{n} ((1-g(X_i))(1-\nu_i) + g(X_i)\nu_i) | X_1^{n-1} \right] \right] f(\nu) =$$

$$\sum_{\nu \in \{0,1\}^n} \mathbf{E}_{X_1^{n-1}} \left[ \left( \prod_{i=1}^{n-1} ((1-g(X_i))(1-\nu_i) + g(X_i)\nu_i) \right) \mathbf{E}_{X_n} [(1-g(X_n))(1-\nu_i) + g(X_n)\nu_i | X_1^{n-1}] \right] f(\nu) =$$

$$\sum_{\nu \in \{0,1\}^n} \mathbf{E}_{X_1^{n-1}} \left[ \left( \prod_{i=1}^{n-1} ((1-g(X_i))(1-\nu_i) + g(X_i)\nu_i) \right) ((1-b_n)(1-\nu_i) + b_n \nu_i) \right] f(\nu) = \ldots$$

$$\sum_{\nu \in \{0,1\}^n} \left( \prod_{i=1}^{n} ((1-b_i)(1-\nu_i) + b_i \nu_i) \right) f(\nu) = \mathbf{E}_{Z_1^n} [f(Z_1^n)].$$

$\square$

**Theorem 5.** *For any fixed hyper-prior distribution $\mathcal{P}$ with probability at least $1 - \delta$ the following holds uniformly for all hyper-posterior distributions $\mathcal{Q}$:*

$$\widetilde{\mathrm{er}}(\mathcal{Q}) \leq \widehat{\mathrm{er}}(\mathcal{Q}) + \frac{1}{(n-1)\sqrt{m}} \mathrm{KL}(\mathcal{Q} \times Q_2 \times \cdots \times Q_n || \mathcal{P} \times P_2 \times \cdots \times P_n) + \frac{(n-1) + 8\log(1/\delta)}{8(n-1)\sqrt{m}}, \tag{9}$$

*where $P_2, \ldots, P_n$ are some reference prior distributions that should not depend on the training sets corresponding to subsequent tasks. In particular, it can be just one prior distribution $P$ fixed before observing any data, or posterior distribution corresponding to the previous task, ie $P_i = Q_{i-1}$.*

*Proof.* By applying KL-inequality we obtain:

$$\widetilde{\mathrm{er}}(\mathcal{Q}) - \widehat{\mathrm{er}}(\mathcal{Q}) \leq \frac{1}{\lambda} \Big( \mathrm{KL}(\mathcal{Q} \times Q_2 \times \cdots \times Q_n || \mathcal{P} \times P_2 \times \cdots \times P_n) +$$

$$\log \mathbf{E}_{A \sim \mathcal{P}} \mathbf{E}_{h_2 \sim P_2} \ldots \mathbf{E}_{h_n \sim P_n} \exp \Big( \frac{\lambda}{n-1} \sum_{i=2}^{n} \Big( \mathbf{E}_{(x,y) \sim D_i} \ell(h_i(x), y) - \frac{1}{m} \sum_{j=1}^{m} \ell(h(x_j^i), y_j^i) \Big) \Big) \Big).$$

Due to independence of any prior $P_i$ and consequent sample sets $S_i, \ldots, S_n$, we obtain that:

$$\mathbf{E}_{S_1, \ldots, S_n} \mathbf{E}_{A \sim \mathcal{P}} \mathbf{E}_{h_2 \sim P_2} \ldots \mathbf{E}_{h_n \sim P_n} f_2(h_2, S_1) \cdot \cdots \cdot f_n(h_n, S_n) =$$
$$\mathbf{E}_{A \sim \mathcal{P}} \mathbf{E}_{S_1} \mathbf{E}_{h_2 \sim P_2} \mathbf{E}_{S_2} f_2(h_2, S_2) \ldots \mathbf{E}_{h_n \sim P_n} \mathbf{E}_{S_n} f_n(h_n, S_n),$$

where

$$f_i(h_i, S_i) = \frac{\lambda}{n-1} \Big( \mathbf{E}_{(x,y) \sim D_i} \ell(h_i(x), y) - \frac{1}{m} \sum_{j=1}^{m} \ell(h_i(x_j^i), y_j^i) \Big). \tag{10}$$

Due to Hoeffding's lemma, boundness of the loss and the fact that training samples are i.i.d., the following holds:

$$\mathbf{E}_{S_n} f_n(h_n, S_n) \leq \exp \Big( \frac{\lambda^2}{8(n-1)^2 m} \Big). \tag{11}$$

Therefore:

$$\mathbf{E}_{S_1,\ldots,S_n} \mathbf{E}_{A\sim\mathcal{P}} \mathbf{E}_{h_2\sim P_2} \ldots \mathbf{E}_{h_n\sim P_n} f_2(h_2,S_12) \cdot \cdots \cdot f_n(h_n,S_n) \leq \exp\left(\frac{\lambda^2}{8(n-1)m}\right). \quad (12)$$

By using Markov's inequality and setting $\lambda = (n-1)\sqrt{m}$ we obtain the statement of the theorem.
$\square$

The KL-term in the above theorem can be simplified:

$$\mathrm{KL}(\mathcal{Q} \times Q_2 \times \cdots \times Q_n || \mathcal{P} \times P_2 \times \cdots \times P_n) = \mathrm{KL}(\mathcal{Q}||\mathcal{P}) + \sum_{i=2}^{n} \mathbf{E}_{A\sim\mathcal{Q}} \mathrm{KL}(Q_i||P_i).$$