[Reviews · NeurIPS 2015]

Submitted by Assigned_Reviewer_1

This paper presents a theoretical analysis of Lifelong learning setting which goes beyond the traditional task i.i.d assumption. Two settings are considered: stationary environment with non-independent tasks and non-stationary environments.

I liked the paper, it is written well and I feel I was able to follow it even though I am not an expert in PAC-Bayesian analysis. I did miss a few concrete examples, particularly at the introduction level, motivating the two possible learning settings and relating them (or contrasting) to existing lifelong learning approaches if possible.

A synthetic data example was given for the non-stationary environments, it would have been nice to add an example for the stationary setting as well.
Summary: I think this paper adds an important theoretical analysis to the Lifelong learning setting. I feel the paper would have benefited from a few concrete examples.

Submitted by Assigned_Reviewer_2

The paper presents some extensions to the Pentina and Lampert's PAC-Bayesian analysis of "Lifelong Learning" problems (ICML 2014) , where a learner must adapt to various tasks exploiting knowledge from previously seen ones. The main contributions are risk bounds dedicated to two scenarios where the observed task are not sampled independently from each other. Roughly speaking, the first scenario share similarities with domain adaptation (albeit the risk bound is given on an average of all possible domains, instead of on a specific target domain) and the second is quasi-identical (up to my knowledge) to distribution drift.

In the first setting (Section 3), the authors cleverly reuse Ralaivola et al.'s chromatic PAC-Bayesian theory to represent dependencies between tasks. However, this result alone let me unsatisfied. I wonder to which extent this result can be useful to the ambitious "lifelong learning" problem the authors are interested in. At the end of Section 3, the authors open the way to use the result as a "quality measure" benchmark or to the conception of a learning algorithm. If provided, such empirical study could be a good way to enhance this work.

The second setting (Section 4) is very similar to the "distribution drift" scenario. I would like the authors to compare their results with other ones in this area (e.g., Mohri and Medina's "New Analysis and Algorithm for Learning with Drifting Distributions", ALT 2012). I also think that this setting must be studied more deeply. Contrary to the author claim, I think that the assumption of Equation (18) is very restrictive: it is unlikely that *all* algorithms of a considered family would have *exactly* the same error on every two consecutive tasks. This assumption should be relaxed to become realistic (as in Mohri and Medina 2012). Moreover, the algorithm should be able to consider more than the two previous tasks.

Finally, throughout the paper, I wondered why the authors choose this specific formulation of PAC-Bayesian theorems. While Theorem 1 tradeoff is very easy to understand, the obtained bound is less tight than the bound of McAllester ("PAC-Bayesian stochastic model selection", Mach Learn 2003), where the complexity term appears under a square root, while being "explicit" (see also the slightly tighter version of Germain et al.'s "Risk Bounds for the Majority Vote...", JMLR 2015). Consequently, I presume that the further bounds for non-i.i.d. tasks are not as tight as they can be. I think this must be discussed in the paper.

Some typos and minor comments: - There is a small error in Theorem 1 proof: When applying Hoeffding's lemma (Equation (3) of the Supplementary Material), there should be a "4" before lambda^2, because the subtraction between losses lies in [-1,1]. I also think that the same error appears at Equations (11), (25) and (26) of the main paper. - Line 89: The dot is misplaced - Supplementary Material, Line 78: Fro -> For - For the reader's benefit, it is preferable to cite the more recent version of Ralaivola et al.'s "Chromatic PAC-Bayes bounds for non-iid data" (JMLR 2010 instead of AISTATS 2009) - I personally prefer when the definitions, theorems, lemmas, etc share the same counter.
Summary: The paper is clearly written and contains interesting theoretical results. However, each of the two studied scenarios deserves to be studied more deeply to be published.

Author Feedback
Author rebuttal: We thank the reviewers for their helpful comments.

Assigned_Reviewer_2:

> "the paper would have benefited from a few concrete examples"

We will extend the discussion and include more examples.
Note that the non-i.i.d. setting is very common, e.g.
the active users of speech recognition software are
not i.i.d. sampled of time-zone, etc.

> "example for the stationary setting as well"

see lines 220/221: the setup (in particular the algorithm) for such
experiments would be identical to [Pentina 2014], except with dependent
instead of independent tasks for training.

Assigned_Reviewer_3:

> "I wonder to which extent this result can be useful to the ambitious "lifelong learning" problem" ?

The setting generalizes the setting of [Pentina 2014], which already targeted the lifelong learning
situation. We believe it is a significant generalization, since for real-world learning tasks, an
i.i.d. assumption is unrealistic (see the above time-zone example).

> empirical study

see Reviewer 2.

> relation to "distribution drift"

Both setting have commonalities, but they differ in important aspects:

* in distribution drift, e.g. [Mohri and Medina, 2012], one observes a sequence of examples
from a time-varying data distribution and bounds the error of a hypothesis on future samples
based on its performance at previous time steps.

* in our setting, we observe sample sets corresponding to tasks from a time-varying tasks environment.
At any time step, we learn a new predictor (that might be very different from the one of the previous
time step). Samples from previous tasks act as context/prior, not as training data, and our bound
quantifies the performance not of a single hypothesis, but of a transfer algorithm.

This difference is gives the bounds different characteristics:

* In distribution drift, the bound contains a term that grows linearly with the number of samples.
The tightest bound is achieved by disregarding samples from too far in the past. This makes sense,
since if the underlying data distribution changes, a single hypothesis cannot be expected to work
well in the distant past as well as in the future.

* In lifelong setting, we expect the performance to get better the more tasks we observe.
This is reflected in the bound by the terms decreasing with n. Note that his is only possible
because we bound the quality of transfer algorithms, not fixed hypotheses.

A final difference lies in the used assumptions:

* Distribution drift assumes the changes between steps to be small, but otherwise arbitrary,
i.e. a form of worst case assumption.

* We do not assume that changes between task environments are small, but "consistent",
i.e. past experience can be used to identify a transfer algorithm that can compensate
for the changes.
While this is indeed a restriction, we believe it is not as strict as it sounds: it is
not required that all algorithms have exactly the same error on every two consecutive
tasks, as the statement is only in expectation.

We agree, however, that it would be interesting and promising to relax the assumption
to allow for (small) deviations, along the lines of the distribution drift work.
Thank you for the suggestion.

> the algorithm should be able to consider more than the two previous tasks

It would be possible to give the transfer algorithm access to a larger
(but fixed) amount of tasks.

> specific formulation of PAC-Bayesian theorems

We chose this bound mainly to keep the expressions comparable with [Pentina 2014].
Also, we like that it allows the combination of multiple bounds without technical
complications and gives rise more directly to practical algorithms than e.g.
the square-root variant or the 'small-kl' bounds, even if these might be tighter
in some situations.
We'll add an explanation of our choice and the relation to tighter bounds in the
manuscript.

> minor comments:

thank you, we will fix these.

Assigned_Reviewer_4, Assigned_Reviewer_5, Assigned_Reviewer_6:

thank you for your comments.